# Transcriptomic and Proteomic Analysis of Drought Stress Response in Opium Poppy Plants during the First Week of Germination

**DOI:** 10.3390/plants10091878

**Published:** 2021-09-10

**Authors:** Kristýna Kundrátová, Martin Bartas, Petr Pečinka, Ondřej Hejna, Andrea Rychlá, Vladislav Čurn, Jiří Červeň

**Affiliations:** 1Department of Biology and Ecology, Faculty of Science, University of Ostrava, Chittussiho 10, 710 00 Ostrava, Czech Republic; tynakundrat@post.cz (K.K.); martin.bartas@osu.cz (M.B.); petr.pecinka@osu.cz (P.P.); 2Department of Genetics and Agricultural Biotechnology, Faculty of Agriculture, University of South Bohemia, Studentská 1668, 370 05 České Budějovice, Czech Republic; hejna@zf.jcu.cz; 3Research Institute of Oilseed Crops, OSEVA PRO. Ltd., Purkyňova 10, 764 01 Opava, Czech Republic; rychla@oseva.cz

**Keywords:** opium poppy, *Papaver somniferum*, drought response, transcriptomics, proteomics, dehydrins, gene expression, plant stress

## Abstract

Water deficiency is one of the most significant abiotic stresses that negatively affects growth and reduces crop yields worldwide. Most research is focused on model plants and/or crops which are most agriculturally important. In this research, drought stress was applied to two drought stress contrasting varieties of *Papaver somniferum* (the opium poppy), a non-model plant species, during the first week of its germination, which differ in responses to drought stress. After sowing, the poppy seedlings were immediately subjected to drought stress for 7 days. We conducted a large-scale transcriptomic and proteomic analysis for drought stress response. At first, we found that the transcriptomic and proteomic profiles significantly differ. However, the most significant findings are the identification of key genes and proteins with significantly different expressions relating to drought stress, e.g., the heat-shock protein family, dehydration responsive element-binding transcription factors, ubiquitin E3 ligase, and others. In addition, metabolic pathway analysis showed that these genes and proteins were part of several biosynthetic pathways most significantly related to photosynthetic processes, and oxidative stress responses. A future study will focus on a detailed analysis of key genes and the development of selection markers for the determination of drought-resistant varieties and the breeding of new resistant lineages.

## 1. Introduction

Drought is one of the main abiotic stresses that limits crop growth and agricultural productivity [1]. Climate change represents a significant threat to the sustainable development of agriculture; thus, for decades, breeders have attempted to develop more stress-resistant cultivars by exploring the physiological and biochemical processes of drought tolerance in plants [2]. In recent years, due to irregular rainfall, the frequency and severity of droughts has increased. Almost every year, water deficiency affects crop yields on some part of the earth [3]. Therefore, it is crucial to understand the mechanism of drought stress tolerance in plants, in order to improve crop productivity under stress conditions [4]. 

Water deficiency affects various physiological and biochemical processes in plants. The following processes involve plant growth and development [5]: photosynthesis, osmotic homeostasis [6], ion absorption and transport, chlorophyll synthesis, respiration, and carbohydrate metabolism, thereby, inhibiting plant growth and reducing yield [7,8]. During drought stress, plants respond with a number of effective self-protection mechanisms. It was previously shown that there are a series of changes in enzyme activity, cell wall metabolism, endogenous hormones, and the response to reactive oxygen species (ROS) in dealing with drought and dehydration [9]. 

Plants also adapt to drought stress by inducing the expression of specific drought tolerance genes [10,11]. One of the specific protective compounds aiding with drought stress are dehydrins [12], which are important protein chaperones [13]. They belong to late embryogenesis abundant (LEA) proteins, which are highly hydrophilic, and their structural analysis implies that they are intrinsically disordered proteins (IDPs) [14]. They stabilize the cell and protect the tissues from water loss [15]. Many studies show that dehydrins are synthesized under abiotic stress in different plants, such as wheat [16], barley [17], chickpea [18], Norway spruce [19], and clover [20]. 

*Papaver somniferum* (opium poppy) is a famous producer of analgesic and narcotic compounds. Due to their analgesic properties, pharmaceutical opiates are universally applied for relieving pain caused by cancer, surgery, and wounds [21]. The opium poppy is also the source of narcotics, such as opium and heroin [22]. Due to this, many countries forbid or strictly control the cultivation of *Papaver somniferum* and the use of opium products through legislation [23]. Nonetheless, *Papaver somniferum* is cultivated as an annual crop in countries such as China, India, the Czech Republic, and Turkey. The main reason for the legal growth of the opium poppy is its valuable oil seed. The seeds are used mostly for bakery products and for oil processing [24]. Poppy seeds contain up to 50% oil and Indian cultivars have high levels of oleic and linoleic acids [25]. Poppy seed oil appears to be of good quality for human consumption since it is generally rich in polyunsaturated fatty acids [26]. In Europe, poppy seeds are mostly used for confectionery. Furthermore, they are a source of drying oils, used for the manufacture of paints, varnishes, soaps, and in food and salad dressing [27].

The identification of genes in non-model plants that lack a reference genome could be achieved by de novo RNA sequencing (RNA-seq). This powerful, high-throughput sequencing technology can be widely applied to identify target genes and pathways related to stress responses. It allows quantitative transcriptomic profiling and the discovery of virtually all expressed genes in plant tissues under abiotic stress [28,29]. However, many gene products are subject to post-translation modification, which cannot be detected by a transcriptomic analysis [30]. Improvements in high-throughput liquid chromatography-tandem mass spectrometry have made proteomics a powerful tool. Studies into protein expression levels, post-translational modifications, and protein–protein interactions have led to a more comprehensive understanding of cell metabolism and other processes which occur at the protein level [31] and thus proteomics can effectively supplement transcriptomic profiling.

In this study, transcriptome and proteome sequencing were performed on two *Papaver somniferum* genotypes with each exhibiting a different phenotypic response to drought stress under both control and drought stress treatments after 7 days of growth, as during the first stages of germination opium poppy plants are most vulnerable to drought and other kinds of biotic and abiotic stress [32]. Dehydrins were analyzed as potentially key proteins in protecting plants from water deficiency in the early germination stage when poppy plants are most vulnerable to drought. In addition to dehydrins, the possible biological functions of the resulting differentially expressed genes (DEGs) and differentially expressed proteins (DEPs) were also assessed under stress conditions. The interaction and expression relationship between these DEGs and DEPs were also explored, along with their potential effect on *Papaver somniferum* under drought stress. This study should add to and enhance transcriptomic and proteomic resources, which will facilitate future research into the discovery of novel genes and provide a reference for the further understanding of the regulatory networks in *Papaver somniferum* when under drought stress.

## 2. Results

### 2.1. Varieties of Papaver Somniferum

Two contrasting varieties (Extaz, Prevalskij 133) were used in this study and were selected based on their differing responses to drought stress during germination. The average germination in drought condition was 88 % for the Extaz variety, and only 12 % for the Prevalskij 133 variety. Thus, the Extaz variety was determined as drought resistant, and the Prevalskij 133 variety was determined as drought sensitive (detailed results can be found in Appendix A).

### 2.2. Gene and Protein Expression of Dehydrins

Dehydrins (DHNs), which are members of the LEA protein family, are reported to be crucial factors in plant response to water deficiency [33]. Hence, at first, we focused on DHNs as the key protein family. We studied the expression of DHNs at the transcriptomic and proteomic levels. In the Extaz variety there were not detected dehydrins that would be significantly differently expressed. On the other hand, in the variety Prevalskij 133 there were found 4 DHNs differently expressed on the protein level between control and drought stress treatment. These results are shown in the Table 1.

*De novo* RNA sequencing was used for transcriptome analysis of *Papaver somniferum*. There were six types of DHNs identified in the transcriptome, and just two types of DHNs were significantly differentially expressed (*p*-value < 0.05) between drought and control conditions (in Prevalskij 133 variety). (Table 1). Table 1 shows results of differentially expressed dehydrins between control and stress treatment in the variety Prevalskij 133. A positive number in log2 fold change column means that the DHNs were more expressed in the control conditions compared to drought stress conditions.

Furthermore, we studied the differential expression of DHNs at the proteomic level. There were four types of DHNs detected in the proteome, and all of them were significantly differentially expressed (*p*-value < 0.05) in response to water deficiency. Table 1 shows results of DHNs frequency between control and stress treatment in the variety Prevalskij 133. A negative number in the log2 fold change column means that the DHNs were more abundant in the drought stress treatment in comparison to control conditions in the variety Prevalskij 133.

In summary, the changes in expression of DHNs, which are most commonly recognized as drought stress response genes, were much lower than we hypothesized (Table 1). Therefore, our following analyses were focused to other genes and proteins related to drought stress. 

### 2.3. Identification of DEGs and DEPs

The analysis revealed 233 DEGs in the Extaz variety (the gene expression changed between the control condition and drought stress treatment). Among them, 122 DEGs were up-regulated in the drought stress treatment. In the Prevalskij 133 variety, there were 1259 DEGs between the control and drought stress treatments. Among them, 576 DEGs were up-regulated in the drought stress treatment.

In addition, there were 1386 DEGs between the Extaz and Prevalskij 133 varieties in the drought stress treatment. Among them, 752 DEGs were up-regulated in the Prevalskij 133 variety, and 634 were up-regulated in the Extaz variety. All analyses are based on a comparative analysis with a false discovery rate adjusted *p*-value < 0.05. Relevant genes that were significantly up- or down-regulated, with a further criterion of log2 fold change >3.9, were plotted onto a heatmap (Figure 1A). The greatest difference in DEGs was between the drought-stressed *Papaver somniferum* varieties. It was clearly visible that the Extaz variety responded differently than the Prevalskij 133 variety under drought stress. 

Interestingly, there were no significant DEPs in the Extaz variety (drought resistant) between the control condition and drought stress treatment. In the Prevalskij 133 variety, there were 11 significant DEPs (between the control and drought stress treatment). Among them, 6 proteins were preferentially induced under drought stress conditions. In addition, there were 52 significant DEPs between the Extaz and Prevalskij 133 varieties in the drought stress treatments. Among them, 25 proteins were more abundant in the Extaz variety, and 27 were found in a higher abundance in the Prevalskij 133 variety. All analyses are based on a comparative analysis with an adjusted *p*-value < 0.05. Resulting DEPs were plotted on a heatmap (Figure 1B). 

The presented heatmaps use a hierarchical cluster analysis, and the results underlined different responses of *Papaver somniferum* varieties to drought on gene and protein levels. A hierarchical cluster analysis of DEGs showed that the Extaz and Prevalskij 133 varieties differed mainly in their transcriptomic responses to drought stress (Figure 1A). On the contrary, DEPs showed there were greater differences between varieties than between treatments within each variety, as varieties clustered together (Figure 1B). A detailed analysis of the most interesting genes/proteins can be found in the discussion section. 

### 2.4. Interaction Networks and Biosynthetic Pathways within DEGs and DEPs

In the results of the DEGs with a log2 fold change > 3.9, the main biological function categories were found to be the leucine catabolic process, L-phenylalanine catabolic process, jasmonic acid biosynthetic process, response to high light intensity, response to hydrogen peroxide, alpha-amino acid catabolic process, response to heat, response to chitin, response to a virus, and response to a temperature stimulus. The interaction network of proteins encoded by drought-tolerance-related DEGs is shown in Figure 2. 

As both the constitutive and induced response to stress is known, we chose to lower an adjusted *p*-value < 0.15, that might involve both types of stress response proteins [39]. Then we analyzed the interaction network of drought-tolerance-related DEPs (Figure 3).

In the results for the DEPs, the main biological functional categories were found to be: xylan catabolic process, melatonin biosynthetic process, mannose metabolic process, protein deglycosylation, removal of superoxide radicals, arginine biosynthetic process, glutamin family amino acid metabolic process, phenylpropanoid biosynthetic process, indole-containing compound biosynthetic process, nitrogen cycle metabolic process, protein peptidyl-prolyl isomeration, flavonoid biosynthetic process, aspartate family amino acid metabolic process, dicarboxylic acid metabolic process, and alpha amino acid biosynthetic process.

The analysis of enriched metabolic pathways (Figure 4) showed some congruences between transcriptomic and proteomic levels, e.g., tetrapyrrole binding and photosynthetic response in general. On the other side, transcriptomic response was unique in the protein folding/refolding pathway, and in amylase activity. To the contrary, proteomic pathways were uniquely enriched in responses to oxidative stress, alkaloid metabolism, and others.

## 3. Discussion

During the year 2016, only 20% of the poppy seeds germinated under unfavorable conditions in the Czech Republic. This correlates with a suggestion for best poppy cultivation that underlines the vulnerability of poppy plants in the very early stages of germination [41]. Therefore, we have focused our attention on this critical phase of poppy plants’ development.

Water deficiency has a negative impact on plant growth [42] and leads to the overproduction of reactive oxygen species (ROS), which could seriously impair the normal function of cells [43]. Plants have different tolerance mechanisms to cope with biotic and abiotic stresses [44]. Under stress, plants commonly accumulate small molecule osmotic adjustment molecules, such as soluble sugars, proline, and betaine [45]. Apart from these substances, plants can also induce the expression of DHN genes in response to drought and other abiotic stress [46]. Experiments in vitro suggest that DHNs can stabilize membranes, protect proteins from aggregation, cryoprotect enzymes, protect nucleic acids, scavenge ROS, and can bind small ligands including water, ice crystals, and metal ions, but these findings need in vivo conformation [47]. Drought stress in our study was induced instantly after sowing seeds—accumulation of dehydrins slowly diminishes during germination, which was initially our area of interest, hence we studied the first week of the plant growth. 

In our study of the whole opium poppy transcriptome, six types of DHNs were found. Just three of the six types of DHNs are associated with drought stress (dehydrin ERD10-like, and two dehydrins DHN1-like) according to the literature listed in Table 1. However, just one type of DHN (dehydrin ERD10-like) associated with drought stress in the literature was significantly differentially expressed (adjusted *p*-value < 0.05). At the protein level, four types of DHN were found. All four types of DHNs were significantly differentially expressed (adjusted *p*-value < 0.05) and three of them were associated with drought stress according to the literature listed in Table 1. Dehydrins are proteins that respond to various types of stress (cold, salinity, drought). The functions of the DHNs we found in the transcriptome/proteome are shown in Table 1. However, it is possible that these DHNs may respond to more than one type of stress, for example, dehydrin COR47-like response to drought-stress has not been noticed in the literature. However, the absolute values of the log2 fold changes of gene/protein expression were low. These results suggest that DHNs are not the main protein family which protects *Papaver somniferum* against drought stress in the early stages of germination. Even though the number of differentially expressed genes is significantly higher than differentially expressed proteins, surprisingly most DHNs were found among DEPs and not DEGs. It could be explained by DHNs’ long half-life under stress conditions [48]. As we can see, only some of the dehydrins were found to be up-regulated on the mRNA level, but all of those that were detected on the protein level were more abundant in stress conditions. This might point out their early onset. In stress experiments performed on more developed plants, dehydrin response usually happens under 24 h after treatment [49,50]. As our experiments were carried out on developing plants, we were expecting DHNs even at this timepoint, but most probably other described mechanisms have a major role at this stage. 

Under drought stress conditions, the number of DEGs was lower in the Extaz variety in contrast to the Prevalskij 133 variety. The DEPs were not even found in the Extaz variety after water deficiency. These results suggest that desiccation induces a chaotic reaction in the Prevalskij 133 variety, whereas the Extaz variety reacts to drought stress with a more targeted response. It could be the reason why the Extaz variety is more drought tolerant. Down-regulation of DHNs is rarely seen in drought stress response, but as the expression of DHNs is regulated by different plant hormones, it is expected that the onset of other mechanisms will cause the down-regulation of earlier processes. In experiments on rice, jasmonic acid (JA) caused down-regulation of several DHNs in more plant organs after 12 h [51]. JA regulates different stress responses in plants, including responses to such abiotic factors as cold, salinity, and drought [52].

A summary of the important genes which possibly respond to drought stress was created and is shown on both the heatmap (Figure 1A) and co-expression network (Figure 2). This included the heat-shock protein family (AT5G51440, HSP17.4, HSP17.6II, HSP90.1, Hsp70-2, HSP70, HSP18.2, and CR88), which was expressed in the Extaz variety with the drought stress treatment. Certain proteins from this family encode a group of conserved chaperone proteins that play a central role in the cellular networks of molecular chaperones [53,54,55,56] and folding catalysts [53]. Several heat-shock proteins are involved in drought tolerance [53,54,55,56,57,58] and are found in different organisms including bacteria, plants, and animals [53,54]. The chaperone DnaJ (AT2G20560) also belongs to this protein family and is known as Hsp40. The chaperone DnaJ was up-regulated in the Extaz variety in both conditions, however, the higher change in expression was in the drought stress treatment. The Extaz variety probably increases the abundance of heat-shock proteins during water deficiency and these proteins enhance the plant’s tolerance to drought stress. 

An increased expression of DREB1a was found in both varieties in the control condition. Although the overexpression of the DREB protein family improves tolerance to freezing and dehydration stress in plants [59], Dossa et al. (2016) [60] mention that these genes from the same group could be expressed differently in response to drought stress and, therefore, are thought to have different functions.

Another gene of interest was E3 ubiquitin-protein ligase RZFP34 (AT5G22920), which was mostly expressed in the Prevalskij 133 variety (drought-sensitive) under drought stress conditions. Ding et al. (2015) [61] state that the overexpression of a ubiquitin E3 ligase increases drought stress tolerance in plants.

The gene expression of chlorophyll a-b binding protein 1 (CAB1) was increased in the Extaz variety in the drought stress treatment. Two CAB1 were also found in the proteome analysis. One of these proteins was down-regulated in the Prevalskij 133 variety in the drought stress treatment. The abundance of the second protein was increased in the Prevalskij 133 variety in both drought stress treatment and control conditions. The chlorophyll a-b binding protein 1 belongs to the PSI responsible for chlorophyll-binding [62]. Sarwer et al. (2019), present that CAB1 is down-regulated under drought stress conditions in *Agave sisalana* [63]. However, Zhou et al. (2015) [62] describe an increased abundance of CAB1 in a drought-tolerant variety of apple. The chlorophyll a-b binding protein 1, which is involved in light reaction, responds to drought stress differently. Our results indicate an agreement with these observations.

Furthermore, a summary of the important DEPs was created. The proteins have been chosen similarly to the selection of genes (above). The chalcone-flavanone isomerase family protein (TT5) is expressed in the Extaz variety in both treatments (control and under stress). Nonetheless, the protein was more abundant under the drought stress conditions. Chalcone-flavanone isomerase is a part of the flavonoid pathway. Chalcone synthase is the first enzyme committed in the biosynthesis of all flavonoids and chalcone isomerase is a subsequent enzyme in this pathway [64]. It catalyzes the stereospecific isomerization of chalcone into flavanone. The step for creating chalcone includes chalcone synthase [65], which plays an important role in responding to drought and salinity stresses [66]. Hence, we assume, that chalcone isomerase also responds to drought stress.

Another protein involved in the response to drought stress was the peroxidase superfamily protein (RCI3, AT5G66390), which was mainly expressed in the Extaz variety in the drought stress treatment. RCI3 is a peroxidase, and it is involved in tolerance to dehydration and salt stresses [67,68].

The increased abundance of superoxide dismutase (FSD2) was found in the Prevalskij 133 variety in both treatments, especially during water deficiency. Superoxide dismutase converts superoxide ions, which are created in the absence of sufficient CO_2_ as an ultimate electron acceptor, into less toxic hydrogen peroxide molecules. Therefore, it helps to increase a plant’s tolerance against abiotic stress [69] including drought [70,71].

Additionally, the abundance of cytochrome P450 (CYP97B3) increased in the Prevalskij 133 variety in both conditions. CYP was most expressed under drought stress. The proteins of this gene family catalyze a considerable array of crucial reactions in a plant’s secondary metabolism [72]. Studies [73,74] suggest that cytochrome P450, among other things, is up-regulated under drought stress. Therefore, we assumed that CYP450 plays a role in plant tolerance against drought.

The correlation between the level of mRNA and the level of proteins is minor at first glance, see the Appendix A, where the intersections of DEGs and DEPs are depicted for each variety/condition. As about three times more of total genes than total proteins were detected due to different sensitivity of used methods, there is a chance we have not detected all the overlaps. Differences could be caused by varied post-transcriptional mechanisms involved in turning mRNA into a protein [75,76]. De Sousa Abreu et al. (2009) suggest that ~40% of the variance in protein expression can be explained by changes at the transcript level, and ~60% by other changes [77]. Differences between the transcriptomic and proteomic analyses also may be affected by post-translation modifications [30]. Generally, plant response to abiotic stresses including drought stress is a complex process, when many different pathways are up- or down-regulated in different stages at different timepoints. In our follow-up study, we will focus on the temporal distribution of different drought stress response mechanisms in geminating opium poppy plants. 

## 4. Materials and Methods

### 4.1. Plant Material and Stress Treatments 

In this study, two varieties of *Papaver somniferum* (Extaz and Prevalskij 133) were used. These varieties are different in their tolerance to water deficiency. The Extaz variety is more tolerant to drought stress in contrast to the Prevalskij 133 variety, which is more sensitive. The seeds used in this experiment from these two varieties of *Papaver somniferum* were obtained from OSEVA PRO Ltd. (https://oseva.cz/new/?p=o_nas). 

The plants were grown in a growth chamber under conditions of 23 °C (150 µM photons/cm^2^), and a 16 h/8 h light/dark cycle. Seeds were sown into a 5 cm nursery pot containing 25 g of nutrient substrate for routine management. The photo documentation is enclosed in Appendix A. Drought stress was induced by reducing the watering in certain pots instantly after sowing seeds. Control pots were watered with 5 mL, and pots with stressed plants were watered with 1 mL *per diem*. Each poppy variety (Extaz and Prevalskij 133) and each treatment (control and stressed) were growing in three separate pots. Three poppy plants (including leaves, stem and root) were collected from each pot after 7 days since sowing, washed with distilled water and preserved in RNAlater^TM^ (Sigma-Aldrich, Germany) until total RNA isolation. The length of the whole plant and the length of root were measured as basic physiological parameter, and results can be found in Appendix A.

### 4.2. RNA Extraction and DEGs Screening

The control and drought-stressed plants were used for a transcriptomic analysis. Three plants from each of four conditions (Prevalskij 133 control; Prevalskij 133 drought stress; Extaz control; Extaz drought stress) were used for the isolation of RNA in three replicates. Because the yield of isolated RNA was insufficient for transcriptome analysis, samples were subsequently pooled and, finally, the transcriptome analysis was performed on three pooled isolates obtained from nine plant samples (for each condition). Total RNA extraction of *Papaver somniferum* was conducted using a Monarch^®^ Total RNA Miniprep kit (New England Biolabs Inc., Ipswitch, MA, USA). RNA purity was determined using a NanoPhotometer (Implen, Germany). Only samples which passed the recommended criteria were sent for next generation sequencing.

RNA-seq libraries and sequencing were carried out on a NovaSeq 6000 platform. A FastQC tool [78,79] was used to control the quality of sequencing data. To remove low-quality sequences, Trimmomatic-0.36.5 [80] was used with a Phred quality score threshold of 20. Read counts were calculated using Kallisto (Kallisto quant; Galaxy Version 0.46.2+galaxy0), and DEGs were obtained using limma (Galaxy Version 3.48.0+galaxy1) based on a false discovery rate adjusted (method Benjamini-Hochberg) *p*-value < 0.05.

### 4.3. Protein Isolation and DEPs Screening

After plant growth, approximately 50 mg of tissue from all plants were pooled for each variety and each treatment and isolated by a TRI reagent, according to the manufacture’s protocols (Molecular Research Center, Inc., Cincinnati, OH, USA).

The resulting peptides were analyzed by liquid chromatography–tandem mass spectrometry (LC–MS/MS) performed using an UltiMate 3000 RSLCnano system (Thermo Fisher Scientific, Waltham, MA, USA) online coupled with an Orbitrap Q Exactive HF-X spectrometer (Thermo Fisher Scientific, Waltham, MA, USA). See Appendix A for the full details regarding the analyses and data evaluation.

### 4.4. Protein Functional Network Analysis

We used the STRING web server (https://string-db.org/ accessed on 14 July 2021) [81] with default parameters to investigate whether the selected set of DEGs and DEPs form functionally enriched networks. Disconnected nodes and proteins not connected to the main network were hidden. *Arabidopsis thaliana* was applied as a reference organism. Metabolic pathway analysis was achieved using “ShinyGO v0.66: Gene Ontology Enrichment Analysis + more”, accessed on 18 August 2021 from http://bioinformatics.sdstate.edu/go/, [40], organism *Papaver somniferum*, false discovery rate adjusted *p*-value cutoff equal to 0.05, and maximum of 30 enriched pathways to show.

## 5. Conclusions

To conclude, the transcriptomic and proteomic landscapes provided a detailed view of the mechanisms for drought response and adaptation at distinct biological levels in *Papaver somniferum* in the early stages of germination, where the poppy is most vulnerable to stress. Based on the transcriptomic and proteomic analyses, we identified candidate genes and proteins that likely contribute to the superior performance of *Papaver somniferum* with regard to drought stress.

The results of the DEGs and DEPs analyses showed that removal of superoxide radicals, glutamine family amino acid biosynthetic process, lignin biosynthetic process, response to high light intensity, response to heat, and response to temperature stimulus were involved in the drought response of *Papaver somniferum.*

## Figures and Tables

**Figure 1 plants-10-01878-f001:**
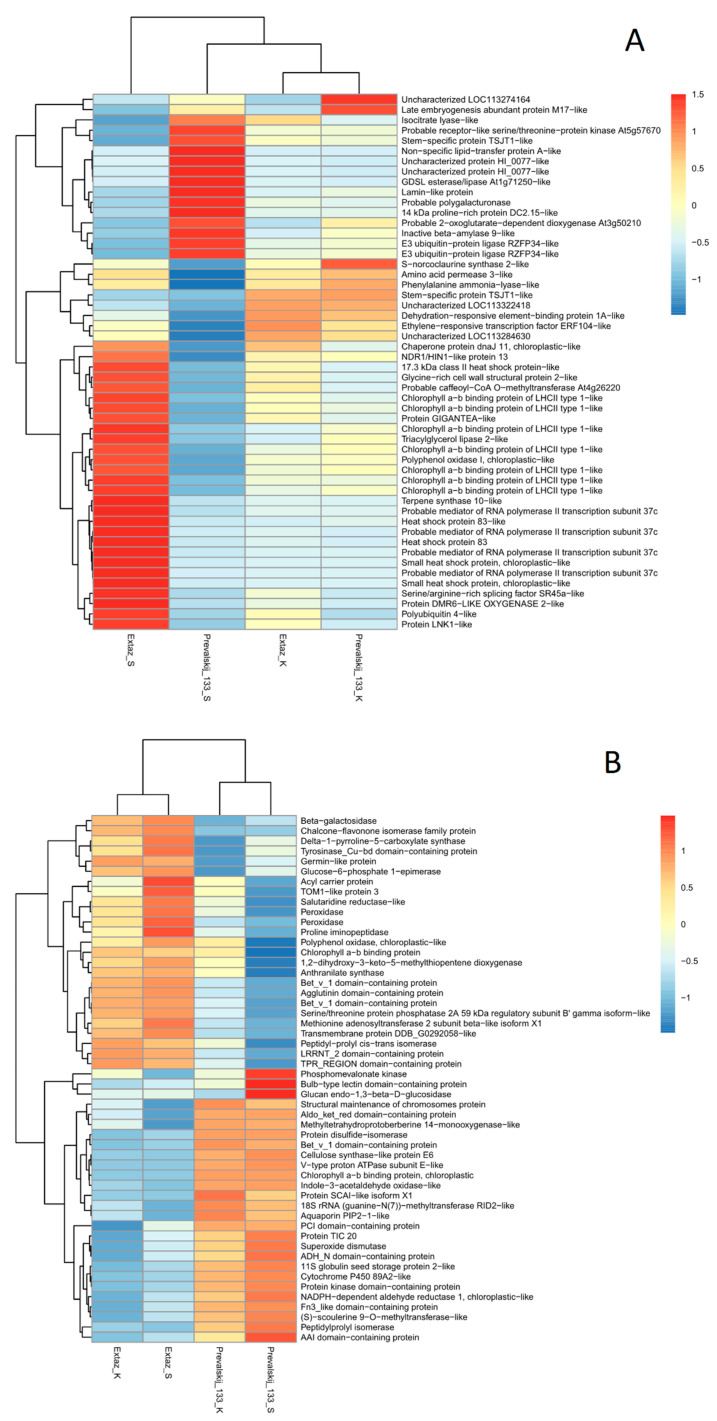
Heatmap showing (**A**) DEGs and (**B**) DEPs in *Papaver somniferum*. The relative differential gene expression was computed using read counts (for genes), or normalized protein intensity signals (for proteins), respectively. The columns marked Extaz_K and Prevalskij 133_K are plants in the control treatment. The columns marked Extaz_S and Prevalskij 133_S are plants in the stress treatment. The red color indicates up-regulated DEGs/DEPs, whereas the blue color indicates down-regulated DEGs/DEPs.

**Figure 2 plants-10-01878-f002:**
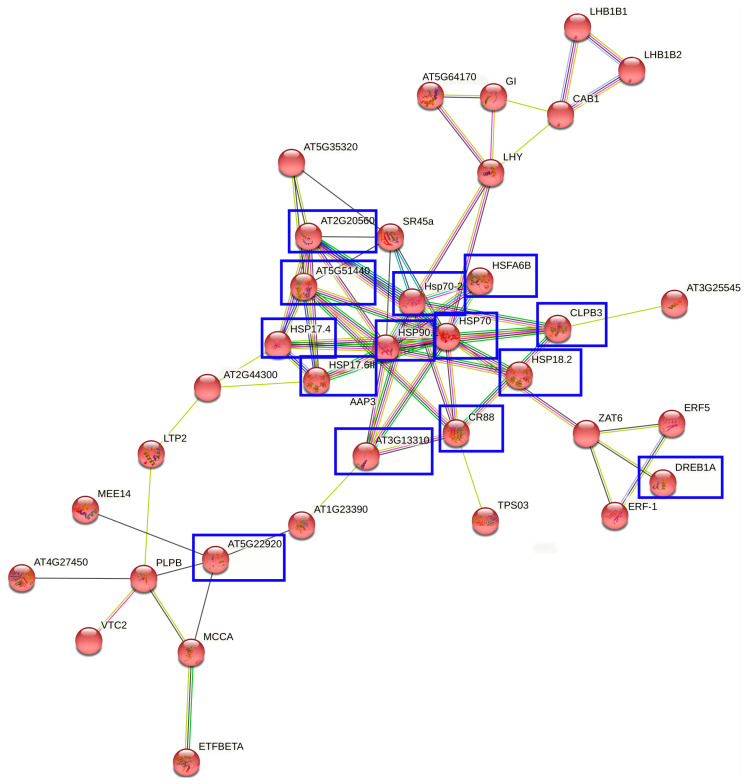
Interaction network of proteins encoded by drought-tolerance-related DEGs. The empty nodes represent proteins of an unknown 3D structure, and the filled nodes signify that some 3D structure is known or predicted. Edges represent protein–protein associations (bright blue and pink color—known interactions, green and dark blue color—predicted interactions, yellow color—text mining, black color—co-expression). Proteins marked in the blue square were significantly associated with drought stress with an adjusted *p*-value < 0.05, and log2 fold change >3.9. Disconnected nodes or proteins not connected to the main network were hidden in the network.

**Figure 3 plants-10-01878-f003:**
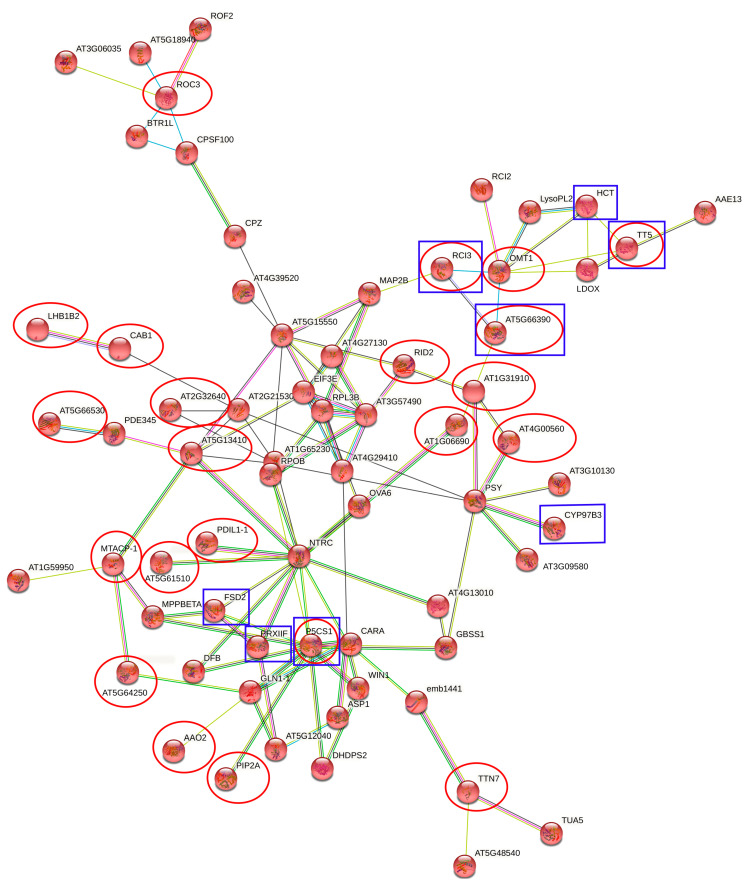
Co-expression network of drought-tolerance-related DEPs. The empty nodes represent proteins of an unknown 3D structure, and the filled nodes signify that some 3D structure is known or predicted. Edges represent protein–protein associations (bright blue and pink—known interactions, green and dark blue—predicted interactions, yellow—text mining, black—co-expression, multicolor—protein homology). Proteins marked in the red circle were those with an adjusted *p*-value < 0.05. Proteins marked in the blue square are associated with drought stress. Disconnected nodes or proteins not connected to the main network were hidden in the network.

**Figure 4 plants-10-01878-f004:**
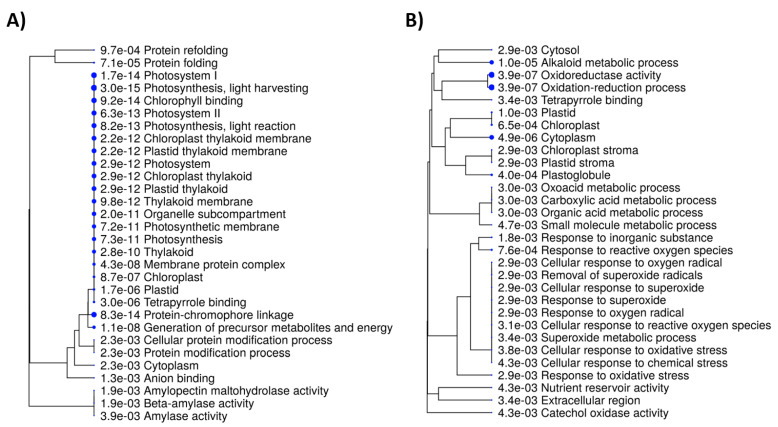
Enriched metabolic pathways under the drought condition in germinating *Papaver somniferum* plants. A hierarchical clustering tree summarizing the correlation among significant metabolic pathways. Pathways with many shared genes were clustered together. Bigger dots indicate more significant *p*-values. (**A**) DEGs, (**B**) DEPs. The plots were produced using ShinyGO webserver (http://bioinformatics.sdstate.edu/go/, accessed on 18 August 2021) [40].

**Table 1 plants-10-01878-t001:** Overview of different types of DHNs found in the transcriptome (light yellow) and proteome (light blue) of the Prevalskij 133 variety (variety sensitive to drought). A negative number in the log2 fold change column means that the DHN was more expressed in the stress treatment compared to the control. Significant log2 fold changes are in bold (*p*-value < 0.05 adjusted to false discovery rate). The names and accession number of DHNs were obtained using BLASTp and the NCBI Database.

Name of DHN	Accession Number	log2 Fold Change	AdjPVal	log2 Fold Change	AdjPVal	Stress Response According to Literature
dehydrin COR47-like	XP_026394044.1	**0.9643**	0.033	**−0.1393**	0.0089	Cold response [34]
dehydrin ERD10-like	XP_026397211.1	**1.0916**	0.0284	**−0.1368**	0.0008	Cold response, dehydration [35]
dehydrin DHN1-like	XP_026393601.1	−0.0677	0.9843	**−0.2548**	0.0005	Drought stress [36]
dehydrin DHN1-like	XP_026432270.1	2.294	0.2181	**−0.1929**	0.0028	Drought stress [37]
dehydrin HIRD11-like	XP_026417500.1	−0.2862	0.7536	-	-	Cold response, UV-B radiation [38]
dehydrin HIRD11-like	XP_026426999.1	−3.4628	0.6364	-	-	Cold response, UV-B radiation [38]

## Data Availability

The sequencing reads generated in this study were deposited in the NCBI Sequence Read Archive (SRA) under accession number PRJNA749648. The proteomic data generated in this study were deposited in the PRIDE Proteomics identification database under accession number PXD027435.

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
