# Peer review of "Transcriptomic and Proteomic Analysis of Drought Stress Response in Opium Poppy Plants during the First Week of Germination"

_plants, 2021, doi:10.3390/plants10091878_

Round 1

Reviewer 1 Report

The manuscript contains original data on the transcriptome and the proteome of Opium poppy plants challenged by water limitation stress during early growth. Two varieties with contrasting drought stress response during germination were compared but no data about this difference are presented nor does reference exist.  It is not clear in Materials and Methods how old were the plants when they were stressed – at germination or post- germinative growth, or cotyledons and rosette leaves alredy have emerged. It is important because the main interest of the authors is about dehydrins and this kind of proteins accumulates in ripening seeds, and then slowly diminishes during germination. The duration of treatment and the effect of the applied stress on water status and plant biomass should be illustrated, probably in Supplementary (I could not open the supplementary files). It is not clear if analyses are on whole plantlets or on different plant organs. In Conclusions a phrase is written “in the early stages of germination”. Please describe clearly the experimental design with more details.

Discussion – the sentence “Water deficiency leads to the overproduction of reactive oxygen species (ROS), which limit plant growth and decrease photosynthesis” should be rewritten as secondary oxidative stress is a consequence of many severe or prolonged stress  conditions and is not the main cause for reduced plant growth and decreased photosynthesis. Discussion is fragmentary and could be more elaborated.

Abstract – the sentence “We found that the transcriptomic and proteomic profiles significantly differ - this is an interesting discovery” is strange as this fact is well known, that is why the authors have studied in parallel transcriptomic and proteomic profiles

In all, in my opinion the manuscript contains significant novelty and should be published after revision.

Author Response

The manuscript contains original data on the transcriptome and the proteome of Opium poppy plants challenged by water limitation stress during early growth.  

Dear Reviewer, thanks for the positive feedback and the time you devoted to reading our manuscript. Our responses to your suggestions are in bold. 

Two varieties with contrasting drought stress response during germination were compared but no data about this difference are presented nor does reference exist.  

Thanks for the fair comment, the contrasting nature of two selected opium poppy varieties was determined based on their differing responses to drought stress during germination. Supporting data about contrasting varieties can be now found in Supplementary material 1. 

It is not clear in Materials and Methods how old were the plants when they were stressed – at germination or post- germinative growth, or cotyledons and rosette leaves already have emerged. It is important because the main interest of the authors is about dehydrins and this kind of proteins accumulates in ripening seeds, and then slowly diminishes during germination. The duration of treatment and the effect of the applied stress on water status and plant biomass should be illustrated, probably in Supplementary (I could not open the supplementary files). It is not clear if analyses are on whole plantlets or on different plant organs. In Conclusions a phrase is written “in the early stages of germination”. Please describe clearly the experimental design with more details. 

Thank you, we did extensive revisions of the Materials and Methods section, to detailly describe the whole experimental process, to be better reproducible in the future. The weight of opium poppy could not be detected. The plans were polluted with soil, subsequently purged with distilled water, which affected the final weight of samples. Hence the length of opium poppy stems and roots was measured instead of the weight. We added the measurements to supplementary material 3, and photos of plant material used (supplementary material 1). 

Discussion – the sentence “Water deficiency leads to the overproduction of reactive oxygen species (ROS), which limit plant growth and decrease photosynthesis” should be rewritten as secondary oxidative stress is a consequence of many severe or prolonged stress conditions and is not the main cause for reduced plant growth and decreased photosynthesis. Discussion is fragmentary and could be more elaborated. 

Thanks, we reformulated the sentence and made substantial improvements to the Discussion section. 

Abstract – the sentence “We found that the transcriptomic and proteomic profiles significantly differ - this is an interesting discovery” is strange as this fact is well known, that is why the authors have studied in parallel transcriptomic and proteomic profiles 

Yes, we agree with you, the sentence was accidentally formulated, we have rewritten it, to more accurately express our results. 

In all, in my opinion the manuscript contains significant novelty and should be published after revision. 

Thank you for your positive recommendation.  

Reviewer 2 Report

MS plants-1339612, entitled: Transcriptomic and proteomic analysis of drought stress response in Opium poppy plants during the first weeks of germination, shows the differential expression and protein profiles of two opium varieties with apparent contrasting behaviour under drought conditions. MS is apparently well written and the results may be interesting to plant scientists. However, the methodology is largely incomplete, and the results are soundless without further information (i.e. there is no hint into how many independent biological samples are used in the two analyses (transcriptomic and proteomic), therefore, it is not clear how significant these expressions or protein abundance data are. The authors should clearly state how many biological replications are used. They should also clearly explain how long was the water stress imposed. And to how old seedlings. Drought stress experiments are not often done on germinating seedlings, therefore it is needed to clearly state the age of the plants, the length of the treatment, and it would also be advisable to show some of the physiological effects of this stressful treatment. Some other points concern the presentation of figures, first, it will be advisable to use the same order of samples in the first two figures (heatmap representations). Similarly, the string representation should not be entitled “Biosynthetic pathways containing DEG/DEP “. In fact, it is only a representation of a protein interaction network. Authors could easily search for metabolic pathways and use them instead. 

Author Response

MS plants-1339612, entitled: Transcriptomic and proteomic analysis of drought stress response in Opium poppy plants during the first weeks of germination, shows the differential expression and protein profiles of two opium varieties with apparent contrasting behaviour under drought conditions. MS is apparently well written and the results may be interesting to plant scientists. 

We thank the Reviewer for the kind words and the time devoted to reading our manuscript. Our responses to the suggestions are in bold. 

 However, the methodology is largely incomplete, and the results are soundless without further information (i.e. there is no hint into how many independent biological samples are used in the two analyses (transcriptomic and proteomic), therefore, it is not clear how significant these expressions or protein abundance data are. The authors should clearly state how many biological replications are used. They should also clearly explain how long was the water stress imposed. And to how old seedlings. Drought stress experiments are not often done on germinating seedlings, therefore it is needed to clearly state the age of the plants, the length of the treatment, and it would also be advisable to show some of the physiological effects of this stressful treatment. 

Thank you for the suggestion. Investigation of drought stress during the first stages of germination was the main aim of our work, as this is a somewhat novel approach we agree with the reviewer and we did extensive revisions of the Materials and Methods section, to detailly describe the whole experimental process, to be better reproducible in the future. Photos of the plant material are enclosed in Supplementary material 1, and we added measurements of stems and roots (Supplementary material 3). 

Some other points concern the presentation of figures, first, it will be advisable to use the same order of samples in the first two figures (heatmap representations).  

Thank you for this comment, we discussed the use of heatmaps for a long-time prior to publication, and we have decided to present it in this way because it underlines differences in DEGs and DEPs as both heatmaps are performed to cluster not only by rows but also by columns (based on similarity of varieties/conditions). We have extended the explanation of the clustering, but if the reviewer agreed, we would like to keep both figures as they are. 

Similarly, the string representation should not be entitled “Biosynthetic pathways containing DEG/DEP “. In fact, it is only a representation of a protein interaction network. Authors could easily search for metabolic pathways and use them instead. 

Thank you, we have reformulated and renamed these results based on reviewers' suggestions and added new results regarding enriched biosynthetic pathways (Figure 4). We are grateful for this suggestion, as these results significantly improve our paper. 

Reviewer 3 Report

The topic is relevant to the profile of Plants and also hold intriguing findings about differences in the transcriptomic and proteomic profiles.

lines15-16  ’In this research, drought stress was applied to Papaver somniferum (the opium poppy), a non-model plant species, during the first weeks of its germination.’

Please explain, that is the first week of germination so important, that it worth investigating, rather than other developmental periods?

lines 18-20 I think the focus of the abstract needs some chnages. It is indeed an inportant-thogh not entirely new discovery- that the transcriptomic and proteomic profiles may differ, but the identification of a key group of genes/proteins with significantly different expressions relating to drought stress is the major finding of this MS.

Please emphasize this a bit more in the Abstract and in the MS.

lines 218-222 ’However, just one type of DHN (dehydrin DHN1-like) associated with drought stress was differentially expressed with comparative analysis at an adjusted p-value < 0.05. At the protein level, 4 types of DHN were found. All four types of DHNs were differently represented with comparative analysis at an adjusted p-value < 0.05 and 3 of the 4 DHNs were associated with drought stress.’

Please explain why there was only one DHN associated with drought, while at the protein level, 4 types of DHN were found. Or were the DHNs found at the protein level not - or not all - associated with drought? If the latter is the case, please separate the thoughts, since they are not closely related and the second thought does not directly rise from the first.

line 315 Please use sowing instead of planting!

line 320 ’Due to insufficient plant material and a low concentration of isolated RNA, the samples were pooled.’

What do Authors mean by insufficient plant material? A trustworthy investigation cannot be done with insufficient plant material, so please provide a detailed explanation why you decided to continue the experiment when according to the Author’s judgement the plant material was insufficient?

Please provide pictures of the plant material that was used for RNS extraction.

Please explain the reasons for low RNA concentration? At the same time, please provide a more detailed description of sample collection (circumstances, temperature, weight, tools used for sample gathering etc.)

lines 293-298. The last paragraph is confusing in terms of the aims of the work. Did the Authors intend to provide a comparison of the two varieties under drought stress? Or is the emphasis is on post-translational processes? If so, the MS is not detailed enough.

’Differences between the transcriptomic and proteomic analyses may also be affected by post-translation modifications’ This is not new information. The new information of this work is about the differences of the varieties and only partly the fact that transcriptional changes do not predict translational changes.  

Moreover, the results were analysed for genes and for proteins, but a comprehensive analysis was not done. Please try to connect the gene results with the protein results better.

Author Response

The topic is relevant to the profile of Plants and also hold intriguing findings about differences in the transcriptomic and proteomic profiles. 

We thank the Reviewer for the kind comment and the time devoted to reading our manuscript. Our responses to the suggestions are in bold. 

lines15-16  ’In this research, drought stress was applied to Papaver somniferum (the opium poppy), a non-model plant species, during the first weeks of its germination.’ 

Please explain, that is the first week of germination so important, that it worth investigating, rather than other developmental periods? 

We thank the reviewer for this question. In the early stages of germination opium poppy plants (seedlings) are most vulnerable to (not only) drought stress. This is knowledge shared by poppy producers. It is well illustrated by some data from the Czech Republic – only 20% of the poppy seeds start to grow in unfavorable conditions, unfortunately, this information is available only in the Czech language. In the “Best practice poppy growing guide” provided by the Tasmanian government as well as in a similar document provided by the Russian government, the importance of early germination stages is underlined. Unfortunately, these documents are not scientific publications. We have highlighted the importance of early germination stages in the Introduction and Discussion sections and referenced these publications as well as one book, that deals with opium poppy growing strategies. 

lines 18-20 I think the focus of the abstract needs some chnages. It is indeed an inportant-thogh not entirely new discovery- that the transcriptomic and proteomic profiles may differ, but the identification of a key group of genes/proteins with significantly different expressions relating to drought stress is the major finding of this MS. 

Please emphasize this a bit more in the Abstract and in the MS. 

Thank you, we did revisions of the Abstract and discussion section and we tried to more highlight the main aim of our research. 

lines 218-222 ’However, just one type of DHN (dehydrin DHN1-like) associated with drought stress was differentially expressed with comparative analysis at an adjusted p-value < 0.05. At the protein level, 4 types of DHN were found. All four types of DHNs were differently represented with comparative analysis at an adjusted p-value < 0.05 and 3 of the 4 DHNs were associated with drought stress.’ 

Please explain why there was only one DHN associated with drought, while at the protein level, 4 types of DHN were found. Or were the DHNs found at the protein level not - or not all - associated with drought? If the latter is the case, please separate the thoughts, since they are not closely related, and the second thought does not directly rise from the first. 

Thank you for the comment, we have reformulated the section to be easier to understand. All results of DHNs identified in transcriptome/proteome is from the comparison between control and drought stress conditions. Nevertheless, according to literature some types of mentioned dehydrins are associated with other types of stresses (like cold, salinity). This information we wanted to mention. 

line 315 Please use sowing instead of planting! 

Thank you for your accurate term. 

line 320 ’Due to insufficient plant material and a low concentration of isolated RNA, the samples were pooled.’ 

What do Authors mean by insufficient plant material? A trustworthy investigation cannot be done with insufficient plant material, so please provide a detailed explanation why you decided to continue the experiment when according to the Author’s judgement the plant material was insufficient? 

Thank you for this comment. As the plants were small, we had to perform pooling of the samples (isolation from three plants still didn’t provide enough RNA for the transcriptomic analysis). As the first weeks (days) of germination are crucial (as emphasized in the previous answer) we couldn’t expect higher yields of RNA/proteins. We have explained this in detail in the manuscript and did an extensive revision of the whole material and methods section. Some details are in the main manuscript, and some were presented as extra SI. We are preparing a more detailed follow-up study with more plants in more timepoints, but we wanted to present these results as a short communication, to allow other researchers to benefit from it. 

Please provide pictures of the plant material that was used for RNS extraction. 

Konec stránky 

We have added them in Supplementary materials 1 and 3. 

Please explain the reasons for low RNA concentration? At the same time, please provide a more detailed description of sample collection (circumstances, temperature, weight, tools used for sample gathering etc.) 

Thank you for the comment, this is again the consequence of the very low amount of RNA isolated from the germinating plants. A detailed description of the samples collection is now provided in Materials and Methods. 

lines 293-298. The last paragraph is confusing in terms of the aims of the work. Did the Authors intend to provide a comparison of the two varieties under drought stress? Or is the emphasis is on post-translational processes? If so, the MS is not detailed enough. 

Thank you for this question. Even though our focus is on the comparison of two varieties in stress conditions, we wanted to comment on seeming differences between results from transcriptomic and proteomic analyses. We have added a couple of lines to the discussion section that deal with this and your last comments. 

’Differences between the transcriptomic and proteomic analyses may also be affected by post-translation modifications’ This is not new information. The new information of this work is about the differences of the varieties and only partly the fact that transcriptional changes do not predict translational changes.  Moreover, the results were analysed for genes and for proteins, but a comprehensive analysis was not done. Please try to connect the gene results with the protein results better. 

We thank the reviewer for this point. It was difficult to perform such analysis, as the numbers of detected genes and proteins were significantly different (about three times as many expressed genes were detected than proteins). When we have looked at overlapping genes and proteins, that were significantly differentially expressed among treatments (and/or varieties) we found only a handful of overlaps. We think that this can be due to different sensitivities of transcriptome/proteome analysis (almost three times as many total genes than proteins were detected). We have added Venn diagrams showing these results as Supplementary material 2 and commented on this in the discussion section.  

Round 2

Reviewer 2 Report

In the revised MS plants-1339612 several sections have been largely improved. However, there are still some aspects that should be polished before publication. There are several minor interpretation and writing mistakes that should be corrected. But, the most challenging is the interpretation of the results, mainly in the comparison among the two contrasting varieties. As an example, when authors mention the importance of a gene or protein in the protection against stress, it should be clear that these genes of proteins usually accumulated under stress, then if the results found is downregulation (as it happens with the dehydrins), the result is far from the expected one. This example can be also applied to many of the DEGs and DEPs. Authors should not only interpret their results as DEGs and DEPs, but also consider when these differences in expression levels coincide with the expected results. Same holds true for the comparison among the tolerant and sensitive variety. Authors many times mention that there are more or less differences among control and treated samples, but the correlation with the role of these differences with the level of tolerance in the two varieties is not clearly stated.

According to the minor changes:

Line 17 , two contrast varieties”, I suggest to change to two drought stress contrasting varieties.

Line 22, remove -this confirms processes inflicted by posttranscriptional and posttranslational modifications. I do not believe that the results presented clearly state that posttranscriptional and posttranslational modifications are responsible for the different profiles among transcriptomic and proteomic analyses. It could be consider as a possible reason, but not as a conclusion, and it should not be included in the abstract.

Line 230. “During the year 2016 only 20% of the seeds germinated in unfavorable conditions in 230 the Czech Republic”; I understand that the authors refer to 20% of poppy seeds. Please change to: During the year 2016 only 20% of the poppy seeds germinated under unfavourable conditions in the Czech Republic.

Line 238:  “plants commonly accumulate small molecule osmotic adjustment substances”. Remove substances, replace by “plants commonly accumulate small osmotic adjustment molecules”.

In summary, I would recommend a thoroughly revision of the text, to better explain the differences in response to the stress found in the two contrasting varieties. 

Reviewer 3 Report

Thank you for the thorough revision of the manuscript. 

The extended versions of the materials and methods and the supplementary materials provided by the Authors improved the manuscript. Therefore, I propose the acceptance of the manuscript for publication in Plants-MDPI.

Author Response

Thank you for the thorough revision of the manuscript. 

The extended versions of the materials and methods and the supplementary materials provided by the Authors improved the manuscript. Therefore, I propose the acceptance of the manuscript for publication in Plants-MDPI.

Thank you for your positive recommendation.